# The Impact of Physical Exercise on Oxidative and Nitrosative Stress: Balancing the Benefits and Risks

**DOI:** 10.3390/antiox13050573

**Published:** 2024-05-07

**Authors:** Qing Meng, Chun-Hsien Su

**Affiliations:** 1School of Physical Education, Huaqiao University, Xiamen 361021, China; mq@hqu.edu.cn; 2Sport and Health Research Center, Huaqiao University, Xiamen 361021, China; 3Department of Exercise and Health Promotion, Chinese Culture University, Taipei 111369, Taiwan; 4College of Kinesiology and Health, Chinese Culture University, Taipei 111369, Taiwan

**Keywords:** exercise physiology, oxidative stress, antioxidant mechanisms, hormesis, nutritional antioxidants, exercise recovery, inflammation response

## Abstract

This review comprehensively evaluates the effects of physical exercise on oxidative and nitrosative stress, mainly focusing on the role of antioxidants. Using a narrative synthesis approach, data from empirical studies, reviews, systematic reviews, and meta-analyses published between 2004 and 2024 were collated from databases like PubMed, EBSCO (EDS), and Google Scholar, culminating in the inclusion of 41 studies. The quality of these studies was rigorously assessed to ensure the clarity of objectives, coherence in arguments, comprehensive literature coverage, and depth of critical analysis. Findings revealed that moderate exercise enhances antioxidant defenses through hormesis, while excessive exercise may exacerbate oxidative stress. The review also highlights that while natural dietary antioxidants are beneficial, high-dose supplements could impede the positive adaptations to exercise. In conclusion, the review calls for more focused research on tailored exercise and nutrition plans to further understand these complex interactions and optimize the health outcomes for athletes and the general population.

## 1. Introduction

Physical activity is intimately connected to the body’s management of oxidative and nitrosative stress, crucial processes that involve the production and build-up of reactive oxygen species (ROS) and reactive nitrogen species (RNS). These processes are fundamental for cellular signaling, regulation, and maintaining balance [1,2]. The impact of exercise-induced oxidative stress on the body can vary, producing positive or negative outcomes depending on exercise intensity, duration, and an individual’s health status [3,4]. Regular exercise enhances the body’s antioxidant capabilities and overall redox balance, contributing to numerous health benefits. However, excessive or poorly tailored exercise can increase oxidative damage and impair muscle function.

The relationship between exercise and oxidative stress is complex, influenced by factors such as the exercise’s type, intensity, and duration. Supplementing antioxidants may be beneficial in situations characterized by high oxidative stress or inadequate dietary antioxidant intake. There is a pressing need for further research to fully understand how various exercise protocols affect the management of oxidative and nitrosative stress across different populations. This necessity is driven by the growing understanding of how exercise-induced biochemical responses play pivotal roles in health outcomes. While generally advantageous, physical activity can also provoke oxidative and nitrosative stress, leading to beneficial adaptations or potential cellular damage, contingent on the exercise’s intensity and duration. This study explores the intricate relationship between exercise-induced stress and antioxidant responses, aiming to inform targeted interventions that maximize health benefits across diverse populations.

### 1.1. Importance of Understanding Oxidative and Nitrosative Stress

The role of oxidative and nitrosative stress in the development of diseases such as cardiovascular diseases, neurodegenerative disorders, diabetes, and cancer cannot be overstated. These stresses have a dual nature, with the potential to be damaging at high levels and regulatory at low levels. Reactive oxygen species (ROS) and reactive nitrogen species (RNS) can impact health and disease in complex ways, modulating inflammation, apoptosis, and mitochondrial function. High levels of ROS resulting from oxidative stress have been linked to the pathology of many diseases including metabolic syndrome, atherosclerosis, Alzheimer’s disease, and rheumatoid arthritis [5]. Oxidative stress is also related to various chronic diseases such as cardiovascular diseases, neurodegenerative diseases, infections, and cancer [6]. Excessive ROS production can alter cellular structure and function, leading to aging and chronic degenerative pathologies [7]. However, antioxidant interventions can help mitigate oxidative stress and its effects on cardiovascular health [8]. Antioxidants have been shown to reduce the harmful effects of oxidative stress on the body, helping to promote better health and reduce the risk of disease [9].

### 1.2. Brief Overview of Mechanisms behind Exercise-Induced Stress Responses

Physical exercise profoundly impacts metabolism by increasing the production of reactive oxygen species (ROS) and reactive nitrogen species (RNS). During aerobic activities, the enhanced oxygen consumption of muscles leads to electron leakage from the mitochondrial electron transport chain, elevating the ROS levels [10]. While potentially compromising cellular integrity, these elevated ROS and RNS levels also serve crucial signaling roles that activate antioxidant defenses and other protective mechanisms [11]. In cases where oxidation products exceed the body’s antioxidant defense capabilities, oxidative stress detrimentally affects the muscles’ molecular, structural, and functional integrity [12]. Regular exercise beneficially activates pathways such as erythroid-related nuclear factor 2 (NRF2) and antioxidant-responsive elements (ARE), which enhance cellular antioxidant defenses and help maintain cellular homeostasis and proper mitochondrial function [13].

Incorporating sociodemographic factors like age, gender, ethnicity, and socioeconomic status is essential for understanding the variability in responses to exercise-induced oxidative and nitrosative stress. Age affects the recovery dynamics and ROS production, with younger individuals often having more efficient antioxidant responses than older adults. Gender differences such as the antioxidant effects of estrogen in women influence oxidative stress levels. Genetic predispositions associated with ethnicity can modify how reactive species are metabolized, impacting an individual’s susceptibility or resilience to oxidative stress. Additionally, socioeconomic status plays a role in access to antioxidant-rich diets and safe exercise environments, which are essential for effectively managing oxidative stress.

Engagement in routine physical activities leads to the production of ROS and RNS in the skeletal muscle, resulting in cellular adaptations that include an increase in the production of endogenous antioxidant enzymes. These enzymes enhance the muscle’s capacity to neutralize ROS and RNS, contributing to several health benefits such as improved cardiovascular health, increased insulin sensitivity, and better muscle function [14,15]. Furthermore, exercise induces the production of nitric oxide (NO) via endothelial nitric oxide synthase (eNOS). NO is crucial for vasodilation, regulating blood flow, and delivering nutrients to active muscles. However, excessive NO can react with superoxide to form peroxynitrite, a potent oxidant that induces nitrosative stress, thus necessitating a balance to mitigate health risks [16,17,18].

These insights into the interplay between exercise, oxidative and nitrosative stress, and sociodemographic factors are vital for crafting personalized exercise and dietary interventions. Such tailored recommendations aim to optimize health benefits and minimize oxidative damage, accommodating individual differences in physiology and life circumstances.

## 2. Methodology

### 2.1. Research Design

This general review was formulated to evaluate the effects of physical exercise on oxidative and nitrosative stress, with a particular emphasis on the role of antioxidants. A comprehensive narrative synthesis methodology was employed to integrate and analyze the findings from diverse studies. This section delineates our systematic approach to the literature selection. We concentrated on empirical research, encompassing both quantitative and qualitative studies, to thoroughly investigate the impacts of physical exercise on oxidative and nitrosative stress. Furthermore, our analysis included a review of the existing reviews, systematic reviews, and meta-analyses to provide a holistic examination of the subject matter.

### 2.2. Search Strategy Description

The literature search was confined to studies published within the last two decades (2004–2024) to incorporate the most recent data to reflect contemporary insights, practices, and findings regarding the impact of physical exercise on oxidative and nitrosative stress as well as the mitigative effects of antioxidants. Search terms such as “physical exercise”, “oxidative stress”, “nitrosative stress”, “reactive oxygen species”, “reactive nitrogen species”, and “antioxidants” were strategically combined using Boolean operators (AND, OR) to enhance the specificity of the search outcomes. The databases queried included PubMed, EBSCO (EDS), and Google Scholar to ensure a comprehensive review of the available scholarly literature. The search strategy is presented in Figure 1.

### 2.3. Roles of Authors and Conflict Resolution

Each author was assigned specific roles: initiating the search, screening the articles, extracting data, and drafting the manuscript sections. Disagreements among authors regarding study inclusion were resolved through discussion until consensus was achieved. If consensus could not be reached, a third-party expert in exercise physiology was consulted to make the final decision.

### 2.4. Inclusion and Exclusion Criteria

The initial selection commenced with a corpus of 6531 studies subjected to a stringent multistage filtering process based on explicit inclusion criteria. The initial filtering at the abstract level assessed studies for (i) the employment of quantitative and qualitative research methodologies, (ii) inclusion of human or animal subjects, (iii) open access or peer-reviewed status, (iv) publication dates ranging from 2004 to 2024; and (v) articles penned in English. This phase accommodated various research designs including descriptive, experimental, quasi-experimental, ex post facto, and instrumental studies. Subsequently, the second filtering stage, conducted at the full-text level, specifically targeted studies that presented systematic reviews, meta-analyses, and those providing detailed statistical analyses or were review studies pertinent to the focal topics. Exclusions during this phase were made for document types such as editorials, brief reports, communications, perspectives, concept papers, and opinion pieces. The final filtering phase concentrated on studies examining the impacts of exercise on producing reactive oxygen species (ROS), reactive nitrogen species (RNS), and the corresponding antioxidant responses. Following this comprehensive selection process, 41 studies were deemed suitable for inclusion in this systematic review.

### 2.5. Data Extraction

Data extraction was conducted using a standardized template that captured key elements of each study including descriptions of the study, the participants’ demographics, details of the study design and interventions, results examined, conclusions drawn, practical implications, and a typical ranking of different types of studies. Two reviewers independently executed this process to ensure the precision and dependability of the data extracted.

### 2.6. Quality Assessment

Our research focused primarily on general and narrative reviews, developing a quality assessment framework through an extensive review of existing methodologies for evaluating narrative and systematic reviews. This process identified essential elements to ensure integrity and reliability, adapting and innovating traditional methods to include measures for assessing narrative coherence and argumentative structure—key aspects often neglected in standard tools. We collaborated with research methodology and academic publishing experts to validate and refine our criteria, ensuring relevance and comprehensiveness. Pilot testing on a sample of reviews further confirmed the practical applicability of our framework in discerning quality differences among narrative reviews. The rationale behind our framework was built on four pivotal criteria: clarity of objectives, coherence of arguments, comprehensiveness of literature coverage, and depth of critical analysis. These were designed to clarify the review’s goals, enhance persuasiveness and logical flow, ensure exhaustive literature coverage, and promote critical engagement with the material, advancing scholarly dialogue and improving the utility of narrative reviews in academic communication. This consolidated approach is detailed in Table 1, which outlines the quality assessment of the reviewed studies.

## 3. Exercise-Induced Oxidative Stress

### 3.1. Sources of Free Radicals during Exercise

During physical activity, the body’s metabolic rate increases, leading to higher oxygen consumption and the production of reactive oxygen species (ROS), which escalate oxidative stress potential. This is crucially managed by the body’s antioxidant defense systems in muscle fibers, emphasizing the importance of maintaining redox homeostasis [19]. Mitochondria, essential for ATP production, become significant sources of ROS due to electron leakage from their electron transport chain during increased activity, potentially causing oxidative damage. However, exercise also triggers protective mechanisms such as the superassembly of mitochondrial complex I in rats, which has been shown to reduce lipid peroxidation and mitochondrial oxidative damage, presenting potential therapeutic benefits for metabolic diseases [20,21].

Ischemia-reperfusion injury, which occurs due to temporary blood flow restriction followed by reoxygenation in muscle tissues, contributes to ROS generation, highlighting the role of oxidative stress in exercise-induced muscle damage and the potential mitigating effects of antioxidant supplementation [22]. The inflammatory response to exercise activates immune cells such as neutrophils and macrophages, which produce free radicals essential for muscle repair and regeneration. This inflammatory process is crucial for recovery, particularly after exercise-induced muscle damage, exacerbated by eccentric muscle contractions that cause severe disruptions in sarcomeres and increase inflammation, leading to strength loss and delayed-onset muscle soreness (DOMS). The recovery timeline from such damage varies with the severity of the damage and the muscle’s prior adaptation [23].

Additionally, catecholamines like adrenaline can autoxidize to form free radicals. However, enzymatic reactions and metal ions are more typical sources at physiological pH, with a potential risk of oxidative damage in cardiac cells [24]. Despite the risks of excessive free radical production, moderate levels can initiate adaptive responses that confer health benefits. This underscores the need to understand these mechanisms to optimize antioxidant defenses and muscle health [25]. Regular physical activity boosts the body’s antioxidant capacity, protecting against oxidative stress and illustrating the complex adaptations to exercise-induced oxidative stress. These responses including oxidative stress and inflammatory responses may depend on exercise intensity, influencing the activation of inflammatory cytokines and sirtuin (SIRT) family members [26].

### 3.2. Antioxidant Responses to Acute and Chronic Exercise Conditions

The body combats exercise-induced oxidative stress using enzymatic antioxidants such as superoxide dismutase, catalase, glutathione peroxidase, and non-enzymatic antioxidants including vitamins C and E. These antioxidants are pivotal in neutralizing reactive oxygen species (ROS). While acute exercise transiently enhances these antioxidants for immediate protection, chronic exercise stimulates genetic adaptations that progressively augment the body’s antioxidant capacity, an effect known as hormesis. This phenomenon shows that moderate oxidative stress can fortify the body’s defenses over time [19].

Computational studies have further demonstrated how antioxidants can influence enzyme structures, thereby contributing to disease prevention. This underscores the critical role of antioxidants in controlling cellular ROS levels and maintaining redox homeostasis, highlighting potential avenues for novel therapeutic strategies against ROS-related diseases [27]. ROS production during exercise is driven by factors like mitochondrial activity and inflammation, which are essential for muscle adaptations such as angiogenesis, hypertrophy, and enhanced mitochondrial function, all of which vary according to the type and intensity of exercise [28].

For athletes, it is crucial to balance antioxidant intake with training to optimize both performance and recovery. This balance is vital for activating signaling pathways that regulate muscle hypertrophy, angiogenesis, and mitochondrial biogenesis—fundamental processes that underpin training adaptation and sports performance. Therefore, understanding the equilibrium between ROS production and antioxidant defenses is essential to maximize the exercise benefits and minimize potential oxidative damage. Careful consideration of antioxidant supplementation is necessary to preserve this balance, as excessive intake might disrupt the natural adaptive responses to exercise and impair athletic performance [29]. Table 2 briefly overviews the exercise-induced oxidative stress mechanisms and antioxidant responses.

## 4. Exercise-Induced Nitrosative Stress

### 4.1. Exploration of the Role of Nitric Oxide in Exercise Physiology

Nitric oxide (NO) enhances blood flow, oxygen delivery, and muscle nutrient supply during exercise, improving performance and endurance. NO production increases with physical activity, facilitated by enzymes such as endothelial and neuronal nitric oxide synthase (eNOS and nNOS), which promote vasodilation and muscular perfusion. Notably, the bioavailability of NO decreases with age, impacting exercise capacity. However, regular aerobic exercise combined with dietary supplements like inorganic nitrate, nitrite, l-arginine, and l-citrulline can counteract these age-related declines, mainly benefiting older adults [30,31].

Exercise-induced NO release also benefits vascular function and cardiovascular health including in patients with vascular diseases. Moreover, NO contributes to mitochondrial biogenesis and efficiency, which is crucial for enhancing endurance and reducing fatigue during extended physical activities [32]. While a meta-analysis highlighted a significant boost in mitochondrial oxidative capacity following exercise, results regarding antioxidant capacity and quality were less definitive. This research primarily investigated conditions such as heart failure, peripheral artery disease, and hypertension, where exercise improved mitochondrial function [32]. Therefore, regular physical activity helps sustain NO levels as we age and capitalizes on its benefits for cardiovascular and muscular health, underscoring the integrated role of NO in exercise physiology and health maintenance.

### 4.2. Examination of the Consequences of Excessive Nitrosative Stress Due to Physical Activity

While moderate nitric oxide (NO) levels benefit health and exercise adaptation, excessive NO can cause nitrosative stress, leading to cellular damage. High NO levels react with superoxide anions to form peroxynitrite, a potent oxidant that can damage lipids, DNA, and proteins, altering cell function and integrity. Such stress is linked to various diseases including cardiovascular and neurodegenerative disorders and inflammation. Too much NO can impair recovery, increase muscle fatigue, and reduce performance in exercise. The body has mechanisms like protein nitrosation and denitrosylation to regulate NO effects; however, an imbalance in these processes can lead to health issues, significantly affecting the central nervous system [33,34]. Additionally, the body produces reactive oxygen species (ROS) in different locations including the mitochondria and endoplasmic reticulum, impacting cellular activities and contributing to pathological conditions when mismanaged. Understanding how to control NO and ROS levels is crucial for preventing oxidative and nitrosative stress-related diseases, emphasizing the importance of maintaining a balance to support cellular function and health [35]. For comprehensive information on the mechanisms, benefits, and regulatory strategies of exercise-induced nitrosative stress, please refer to Table 3.

## 5. Balancing Exercise-Induced Stress and Antioxidant Defense

### 5.1. Introduction to the Concept of Hormesis within the Context of Exercise Physiology

Hormesis, a concept in exercise physiology, suggests that moderate oxidative and nitrosative stress from exercise can stimulate beneficial adaptations in the body. This principle is particularly relevant in addressing the increasing prevalence of neurodegenerative diseases in the aging population. Neurohormesis refers to the brain’s adaptive responses to low-level stress, showing the potential to slow and mitigate the impact of neurodegenerative conditions. Herbal compounds like resveratrol, curcumin, and sulforaphane have been identified for their neurohormetic effects, activating stress response pathways to enhance cellular defense against injury and improve immune function. These findings suggest promising avenues for managing neurological disorders and supporting healthy aging [36].

The article also touches on the role of reactive oxygen species (ROS) in maintaining cellular redox balance and how shifts toward oxidative stress can contribute to chronic diseases such as cardiovascular issues and cancer. While direct antioxidant supplementation has shown limited efficacy and potential risks, strategies to induce mild oxidative stress like hormesis could improve the body’s natural defense mechanisms and increase sensitivity to cancer treatments [37]. Key transcription factors, NF-κB and Nrf2, regulate the expression of antioxidant enzymes, demonstrating how low-level stress can enhance health and longevity by triggering protective responses against various stressors [60]. Additionally, the body’s adaptation to environmental, physical, and nutritional stress through hormesis involves epigenetic changes, highlighting the importance of understanding these responses for health and longevity. Such adaptations underline the potential benefits of controlled environmental exposures to activate defense mechanisms against diseases and aging [61]. This holistic view of hormesis underscores the intricate balance between stress and adaptive response, offering insights into leveraging these mechanisms for health benefits.

### 5.2. Strategies for Optimizing the Health Benefits of Exercise While Mitigating Oxidative and Nitrosative Damage

#### 5.2.1. Moderation in Exercise Intensity and Duration

Regular exercise is crucial in triggering adaptive stress responses and bolstering the body’s antioxidant defenses, providing numerous health benefits. However, intense or prolonged physical activity can elevate the production of reactive oxygen species (ROS), leading to oxidative stress in critical tissues like blood and skeletal muscles, which are significant sources of ROS during exercise. This oxidative stress is associated with muscle fatigue and plays a crucial role in muscle adaptation through biochemical signaling. The effects of exercise-induced ROS on health, whether beneficial or harmful, are still subjects of debate. The body’s antioxidant systems, which include enzymes such as superoxide dismutase, glutathione peroxidase, and catalase are essential for neutralizing ROS and maintaining cellular redox balance.

A study involving 25 sedentary adults investigated how various intensities and durations of exercise impacted oxidative stress and antioxidant responses. The study revealed that oxidative stress markers increased following exercise sessions at 50%, 60%, and 70% of peak aerobic capacity across 10-, 20-, and 30-minute durations, showing different effects on antioxidant enzyme activities. These findings suggest that sedentary adults should limit exercise to 70% of their peak capacity to manage oxidative stress effectively [38,39].

Excessive exercise intensity and duration can lead to an overproduction of free radicals like ROS and reactive nitrogen species (RNS), potentially causing cellular damage. While free radicals are a normal part of cellular processes, their accumulation beyond the body’s capacity to neutralize them can damage proteins, DNA, and lipids, contributing to diseases and accelerating aging. Oxidative stress may also trigger mitochondrial dysfunction and impair mitochondrial biogenesis, which is particularly significant in aging. To combat oxidative stress, regular physical activity and antioxidants such as quercetin, resveratrol, and curcumin are recommended for their protective effects. Chronic inflammation and oxidative stress are interconnected and lead to various health issues. Managing oxidative stress through DNA repair mechanisms, antioxidants like glutathione and superoxide dismutase, and engaging in physical activity are crucial for maintaining cellular health. Structured and moderate exercise is vital to mitigating the adverse effects of oxidative stress.

Research also highlights high dropout rates in health clubs, particularly within the initial months, with enjoyment playing a critical role in exercise persistence. A study demonstrated that aligning exercise intensity with personal preferences enhances the enjoyment and supports exercise commitment, positively affecting exercise habits and the desire to continue exercising [62,63]. This insight is valuable for health club professionals in tailoring exercise programs that maximize adherence by aligning with the individuals’ intensity preferences.

Ultimately, this article examines the balance between benefiting from exercise-induced stress adaptations and avoiding oxidative damage, particularly in high-intensity exercise, and its association with muscle damage. It focuses on how free radicals contribute to oxidative stress responses and offers insights for athletes, coaches, and the general population on recovery strategies from intense workouts and the health implications of such activities. The role of antioxidants in aiding the recovery process and mitigating the effects of oxidative stress is emphasized, highlighting their importance in protecting cells and preventing oxidative stress-related diseases [40]. By customizing exercise intensity and duration based on individual capacities, it is possible to optimize the benefits of exercise while minimizing the risks of oxidative stress-related harm. This personalized approach helps individuals maintain a healthy exercise routine that promotes overall well-being and long-term fitness goals.

#### 5.2.2. Nutritional Support

A diet rich in antioxidants from fruits, vegetables, and nuts plays a vital role in boosting the body’s defense against oxidative stress. This topic continues to be explored in research. Antioxidants are essential in protecting the body from damage caused by free radicals. Found naturally in plant-based foods, antioxidants contribute significantly to disease prevention, while industrial antioxidants are used to prevent the oxidation of various products. By neutralizing free radicals and acting as reducing agents, antioxidants are not only integral to supplements in the food industry, but are also researched for their potential in combating heart disease and cancer. The antioxidant properties of foods like apples and grains, which may inhibit cancer growth and disease development, underline the importance of further investigation into how the body absorbs and benefits from these compounds [41]. The article also discusses the critical role of antioxidants in eye health, detailing how different molecules protect against oxidative stress in both the anterior and posterior segments of the eye. This protection is crucial for preventing eye disorders like dry eye disease, cataracts, and age-related macular degeneration, among others. Despite ongoing research, the effectiveness of antioxidant supplementation in avoiding eye diseases remains uncertain, calling for more detailed studies [42]. Moreover, the consumption of vitamin C, especially among athletes, is debated regarding its necessity and impact on performance. Vitamin C is crucial for immune health and combating free radicals, but there is evidence that high doses might impede athletic performance by affecting training adaptations and vascular function. For athletes, consuming less than 1 g per day and prioritizing vitamin C intake through diet over supplements is advised, with the relationship between vitamin C and athletic performance necessitating further research to optimize dosage and timing [43]. Oxidative stress, characterized by the imbalance between pro-oxidants and antioxidants, underscores the importance of managing this balance to prevent cellular damage and dysfunction [62]. Studies have explored using antioxidants like zinc, selenium, and vitamin C to counteract oxidative stress induced by environmental pollutants like cadmium, highlighting the potential of dietary interventions in mitigating oxidative damage. Incorporating a variety of antioxidants through diet and supplements can help maintain a healthy oxidative balance and support overall well-being [64,65].

#### 5.2.3. Adequate Recovery Periods

Adequate recovery periods are essential for mitigating chronic oxidative stress and facilitating the body’s repair mechanisms. Research by Mahendra Wahyu Dewangga and Djoko Pekik Irianto explored how varying exercise frequencies impacted the serum antioxidant levels and muscle damage in male Wistar rats. Their study, which divided the rats into four groups subjected to different exercise frequencies, found that exercising four times a week or daily without sufficient recovery decreased the serum antioxidant levels and increased muscle tissue damage. This underscores the importance of integrating adequate rest days into exercise routines to preserve the body’s antioxidant capacity and muscle health [44].

Active recovery techniques and sufficient sleep are critical in enhancing recovery and boosting antioxidant defenses, effectively counteracting exercise-induced oxidative stress. A specific study examined the fluctuations of oxidative stress biomarkers in ten males performing cycling exercises under hot conditions, analyzing blood samples for oxidative stress indicators. The results did not support the expected increase in exercise-induced oxidative stress, possibly due to the severe heat stress during rest, delayed recovery, or dehydration, with the study’s limited sample size potentially influencing the variability of results. This suggests a need for further research to understand the impacts of oxidative stress when exercising in hot environments [45].

Another study highlighted that strenuous exercise induced beneficial changes in skeletal muscle such as enhanced endurance and strength gains through processes like mitochondrial biogenesis and muscle hypertrophy. However, these benefits were heavily dependent on the recovery phase post-exercise. An imbalance between intense training and inadequate recovery can lead to performance declines, overreaching, and potentially overtraining syndrome (OTS), characterized by prolonged low-frequency force depression (PLFFD) from muscle damage and glycogen depletion. Reactive oxygen, nitrogen species, and inflammatory pathways likely contribute to the development of OTS [46].

Additionally, the autonomic nervous system (ANS) plays a significant role in health by responding to environmental stressors. Imbalances between the sympathetic (SNS) and parasympathetic (PNS) branches of the ANS can contribute to chronic stress and unhealthy lifestyles. The interaction of the ANS with the hypothalamic–pituitary–adrenal axis is crucial in managing immune function, cardiovascular health, oxidative stress, and metabolic imbalances. High-intensity exercise can disrupt this balance, inducing oxidative stress that affects the ANS equilibrium. Maintaining an optimal SNS and PNS balance through exercise is emphasized as it helps reduce oxidative stress and inflammation [66].

Therefore, ensuring the inclusion of appropriate recovery periods in training regimes is vital for optimizing health and performance, highlighting the need to balance exercise intensity with adequate rest to support physiological functions and overall well-being.

#### 5.2.4. Hydration and Electrolyte Balance

Proper hydration and electrolyte balance are critical for optimizing physiological functions and reducing oxidative stress during exercise, especially in conditions that elevate the risk of dehydration such as high temperatures. A study involving 12 healthy men demonstrated that isotonic drinks were more effective than water in protecting muscles from exercise-induced heat stress, highlighting the importance of tailored hydration strategies for enhancing performance and recovery in hot environments [47]. In demanding work settings like those encountered by agricultural workers in Latin America, dehydration and related health complications such as rhabdomyolysis and acute kidney injury are prevalent due to strenuous labor and harsh climate conditions. Despite general water intake and rest guidelines, replenishing electrolytes is critical to maintain proper hydration and prevent electrolyte imbalance. A field trial with sugarcane cutters in Guatemala showed that enhanced electrolyte intake effectively sustained hydration and minimized health risks without compromising productivity [67]. Additionally, data from the National Health and Nutrition Examination Survey indicate that many U.S. adults fail to meet hydration standards, associating insufficient hydration with chronic diseases and increased mortality rates, emphasizing the public health implications of inadequate hydration [48]. This evidence points to the necessity of further research on hydration requirements and the monitoring of hydration status [68]. Thus, ensuring adequate hydration and appropriate electrolyte supplementation is essential for maintaining health, optimizing performance during physical activities, and preventing chronic health issues. For in-depth insights into the comprehensive strategies for optimizing health benefits and mitigating exercise-induced oxidative and nitrosative stress, refer to Table 4 for a detailed overview.

## 6. Nutritional Antioxidants and Exercise

### 6.1. Investigation into How Diet Influences Exercise-Induced Oxidative and Nitrosative Stress

The research underscores the significance of incorporating antioxidant-rich foods into diets to combat exercise-induced oxidative stress. Studies suggest that whole dietary strategies involving foods like dark chocolate, cocoa, oatmeal, and various fruits and juices can enhance the body’s ability to scavenge reactive oxygen species generated during exercise, thereby reducing oxidative stress and inflammation. However, the diverse protocols across these studies call for more standardized research to solidify the evidence supporting the antioxidant benefits of these dietary approaches on exercise-induced oxidative stress [49]. Intense physical activity without adequate recovery can imbalance the body’s free radical production and antioxidant defenses, leading to oxidative damage. However, moderate-intensity exercises such as Taekwondo have been shown to initiate an adaptive response that improves the body’s oxidative balance, emphasizing the importance of exercise type, intensity, and duration in influencing oxidative stress levels and potential cellular damage [50]. Antioxidants from vitamins C and E, selenium, flavonoids, and carotenoids are pivotal in neutralizing harmful reactive species and supporting cellular health during strenuous activities. While supplementation is standard, the correct dosage and timing are crucial to avoid potential pro-oxidant effects and effectively reduce oxidative/nitrosative stress [51].

Diets rich in fruits, vegetables, and whole grains, known for their high antioxidant content, have been linked to improved antioxidant defenses and protection against oxidative stress from exercise. Polyphenols, in particular, found in berries and green tea, may boost antioxidant enzyme activity and lower inflammation and muscle damage markers. The relationship between diet, gut microbiota, and chronic disease development also highlights the role of dietary antioxidants in managing conditions like cardiovascular disease and cancer, promoting plant-based diets like the Mediterranean for their preventive benefits. The discussion extends to the potential of polyphenol supplements in enhancing the recovery and performance of athletes, though the effectiveness and optimal usage of such supplements require further exploration [52,69]. Thus, a balanced diet rich in natural antioxidants and regular exercise is recommended for mitigating oxidative stress and preventing chronic health issues.

### 6.2. Evidence-Based Recommendations for Antioxidant Supplementation to Support Exercise Recovery and Performance

The effectiveness of antioxidant supplementation, especially exceptionally high doses of vitamin C, for athletic performance and recovery presents mixed results. While some research indicates reduced muscle damage post-exercise, other studies highlight neutral or negative impacts on performance and muscle soreness. Consequently, long-term high-dose vitamin C supplementation is not universally recommended due to inconsistent findings and the potential to dampen training adaptations. Athletes are instead encouraged to focus on a diet rich in nutrients for their antioxidant needs [53]. In addition, antioxidants from natural fruit sources, especially those high in polyphenols, are recognized for their anti-inflammatory and antioxidant capabilities, offering a promising approach to muscle recovery and performance enhancement. Strenuous exercise can disrupt the balance between reactive oxygen species (ROS) and antioxidants, leading to delayed-onset muscle soreness (DOMS), which peaks 24 to 72 h after activity. Due to their potent properties, fruit-derived antioxidants have been explored as an effective means to protect muscle cells from ROS-induced damage, showcasing the potential of antioxidant-rich fruit juices in aiding muscle recovery and boosting sports performance. Research on beetroot, grape, and pomegranate juices has shown positive outcomes in minimizing muscle damage and enhancing antioxidant capacity, underlining the role of natural fruit juice supplementation as a viable nutritional strategy for athletes [54]. While the body’s natural training adaptations involve ROS, incorporating specific antioxidants like vitamins C, E, and resveratrol can positively contribute to recovery and athletic outcomes.

Vitamin C and E supplementation: Research on the effects of vitamins C and E on athletic performance and recovery has presented mixed outcomes. Some studies have highlighted the potential of these vitamins to decrease oxidative stress markers and inflammation following intense exercise. However, others caution that high doses might disrupt the body’s natural adjustments to training, possibly limiting gains in endurance and strength. The review delves into the impact of antioxidant and vitamin supplementation across various exercise types including endurance activities and resistance training. Findings generally indicate that such supplementation does not significantly enhance performance, mitigate exercise-induced oxidative stress, or aid recovery. There are concerns that vitamin supplementation could impede the cellular adaptation processes essential for endurance training improvements. The need for further research to clarify the effectiveness and possible downsides of antioxidant and vitamin supplements for athletes is underscored [55,56]. Additionally, a specific focus on the supplementation’s impact on delayed-onset muscle soreness (DOMS) through a review of 14 randomized trials involving 280 participants, mostly young, active individuals revealed that only a few studies had reported a notable reduction in muscle soreness from vitamin C and E supplementation. This inconsistent evidence calls for more comprehensive studies to ascertain the role of these antioxidant vitamins in alleviating DOMS.

Polyphenol supplementation: Polyphenol supplements such as quercetin and resveratrol are gaining attention for their potential to boost endurance performance and enhance the body’s antioxidant defense mechanisms. These compounds might also aid muscle recovery by reducing inflammation. Although widely used by athletes to minimize exercise-induced oxidative stress and speed up recovery, definitive evidence on the effectiveness of dietary polyphenols for athletes remains sparse. This review examines the bioavailability of polyphenols, their efficacy in combating oxidative stress, and their role in improving the antioxidant status and recovery strategies of athletes. Despite indications that polyphenols can bolster antioxidant defenses and mitigate oxidative stress, debates and mixed results persist regarding their efficacy. The need for further research to pinpoint the effects of polyphenols on oxidative stress, antioxidant status in athletes, and the best dosing practices is highlighted. The review also discusses the significance of standardized polyphenol extracts for more reliable research outcomes, alongside the anti-inflammatory properties and contribution to vascular function and recovery enhancement of polyphenols through mechanisms like free radical scavenging and Nrf2 pathway activation. Suggested dosing includes acute supplementation of approximately 300 mg before exercise to boost performance and over 1000 mg daily for several days pre- and post-exercise to aid recovery. However, the complexity of polyphenol absorption and metabolism calls for more in-depth studies to validate these recommendations and fully understand the potential of polyphenols to support rapid recovery between intensive training sessions or competitions [52,57].

Selenium and coenzyme Q10: Selenium and coenzyme Q10 (CoQ10) have been studied for their potential to enhance physical performance and reduce oxidative stress. Selenium is crucial for producing glutathione peroxidase (GPX) and thioredoxin reductase (TxnRd), selenoproteins involved in reducing lipid peroxides and regulating cell death, thus maintaining immune homeostasis and heart function. TxnRd, existing in cytosolic and mitochondrial forms, plays roles in DNA synthesis, angiogenesis, and reducing mitochondrial oxidative stress. These selenoproteins also possess anti-inflammatory properties, potentially suppressing the production of inflammatory mediators. However, the pro-inflammatory effects of some selenoproteins suggest a complex role in inflammation regulation, necessitating further research to clarify selenium’s mechanisms in immunity and inflammation [58]. CoQ10 is vital for the electron transport chain and antioxidant defense, with natural deficiencies linked to aging and exacerbated by certain medications. At the same time, CoQ10 supplementation has shown promise for cardiovascular health; its effectiveness for statin-associated muscle symptoms varies, indicating the need for more research to establish its benefits across different health conditions [59]. This overview highlights the need for further studies to better understand the impact of selenium and CoQ10 on athletic performance, oxidative stress, and health outcomes. For an in-depth analysis of the enhanced understanding of the role of dietary antioxidants in exercise-induced oxidative stress mitigation, refer to Table 5.

## 7. Discussion

This review analyzed the impact of physical exercise on oxidative and nitrosative stress, emphasizing how antioxidant interventions and demographic factors modulate exercise-induced responses.

### 7.1. Modulation of Oxidative Stress through Exercise Intensity

The intricate relationship between exercise intensity and the balance of oxidative and nitrosative stress is crucial for understanding the dual impacts of physical activity on health. Moderate exercise enhances antioxidant defenses, primarily through upregulating enzymes such as superoxide dismutase and catalase [70]. These play critical roles in scavenging free radicals and reducing oxidative damage, thus demonstrating the hermetic effects that promote cellular resilience and longevity [38,39]. This exercise intensity also triggers the NRF2 signaling pathway, a vital regulator of cellular antioxidant mechanisms, thereby boosting the body’s ability to mitigate oxidative stress and protect against cardiovascular and metabolic disorders [25,52,57,60]. Conversely, high-intensity exercise can lead to an overload of reactive oxygen and nitrogen species, overwhelming the antioxidant defense and resulting in cellular damage, muscle fatigue, and impaired recovery, conditions often exacerbated during continuous intense training without adequate rest [26,40,66]. Such scenarios are associated with overtraining syndrome in athletes, characterized by persistent fatigue and reduced performance [29,33,44,46]. Thus, it is critical to design training programs that include proper recovery and nutritional support to moderate the physiological impacts of exercise [34,40,50]. For the general population, incorporating moderate exercise into daily routines significantly enhances health and reduces the risk of chronic diseases related to oxidative stress [40]. This nuanced understanding underscores the importance of personalized exercise regimens, with future research needed to explore these dynamics through longitudinal studies, aiming to clarify the long-term effects of different exercise intensities on oxidative stress and overall health outcomes.

### 7.2. Role of Dietary Antioxidants in Exercise

The interplay between dietary antioxidants and exercise-induced oxidative stress is complex, involving significant biological interactions. The body’s endogenous antioxidant systems respond to moderate exercise by enhancing defense mechanisms to alleviate oxidative damage [37,61]. However, administering exogenous antioxidants like vitamins C and E shows mixed outcomes [71,72]. At the same time, these can diminish oxidative stress markers, and excessive intake may inhibit critical adaptive responses such as mitochondrial biogenesis and the activation of necessary stress response pathways [19,51,55,56,73,74,75,76]. Powers et al. noted that although antioxidants can reduce acute oxidative damage, they might also impede long-term physiological adaptations such as improvements in the body’s natural antioxidant capacity and mitochondrial efficiency, indicating a possible trade-off between immediate oxidative stress reduction and delayed muscular adaptations [19,38]. The effectiveness of antioxidants appears more pronounced when sourced from natural foods, with dietary polyphenols enhancing antioxidant status and supporting recovery and performance effectively when part of a balanced diet, possibly due to the synergistic effects of nutrients within natural food matrices [52,54,57,69,77,78]. These sources like fruits, vegetables, and whole grains deliver complex nutrients that supplements might not replicate [41]. Furthermore, individual factors such as age, gender, fitness level, and exercise type also influence the interaction between dietary antioxidants and exercise-induced oxidative stress; for instance, older adults might need more nutritional antioxidants due to diminished endogenous production, whereas athletes should manage their intake not to counteract essential physiological stress responses [3,12,26,79]. Ongoing research is vital to ascertain the optimal forms, timing, and dosages of antioxidants across various exercise modalities, aiming to enhance the health and performance benefits of exercise without compromising the adaptive responses essential to exercise physiology.

### 7.3. Demographic Variability in Oxidative Stress Response

Understanding how demographic factors such as age, gender, health status, ethnicity, and socioeconomic background influence oxidative and nitrosative stress responses to exercise is crucial for developing personalized health and fitness programs [80,81,82,83,84]. Older adults, who naturally experience declines in antioxidant production and mitochondrial function, may benefit from moderate-intensity exercises combined with antioxidant-rich diets to counteract age-related oxidative stress [85]. Additionally, gender differences, often influenced by hormonal variations like estrogen in women, may necessitate adjustments in exercise intensities or modalities to optimize health outcomes across genders [86]. Individuals with chronic conditions need specially tailored exercise plans to avoid exacerbating oxidative stress, while ethnic and socioeconomic disparities can impact susceptibility to oxidative damage and access to necessary resources, highlighting the importance of public health initiatives aimed at providing equitable health opportunities [87,88,89]. Future research should employ longitudinal studies and personalized medicine principles to thoroughly investigate these demographic distinctions, thereby improving our ability to tailor exercise and dietary interventions to optimize individual health outcomes.

### 7.4. Future Research Directions

Future research should focus on developing precise, personalized guidelines that integrate exercise with antioxidant supplementation, tailored to the specific needs and demographic characteristics of the individuals. This should involve longitudinal studies to monitor long-term physiological adaptations to various exercise intensities and assess the roles of dietary and supplemental antioxidants in diverse populations. Moreover, a deeper exploration into the molecular mechanisms by which exercise impacts redox balance in different tissues will enhance our understanding of how antioxidants influence exercise-induced physiological responses. Integrating advanced technologies such as genomics, proteomics, and metabolomics will provide detailed insights into individual responses to exercise-induced oxidative stress, supporting the development of customized exercise and nutrition strategies. Clinical perspectives on these interventions are essential, particularly in considering how such strategies can be implemented in routine clinical practice to manage or prevent health conditions. Additionally, recognizing the influence of social, psychological, and environmental factors on exercise behaviors and outcomes is crucial for designing effective and culturally sensitive interventions. Collaborative efforts among researchers, healthcare providers, and policymakers are vital to ensure that scientific findings are translated into practical, accessible health applications that promote healthier lifestyles across various communities.

## 8. Conclusions

In conclusion, this review has comprehensively analyzed the impacts of physical exercise on oxidative and nitrosative stress, particularly emphasizing the role of antioxidants. Our findings underscore that exercise generally promotes health by enhancing the body’s antioxidant defenses and redox balance; it can also induce oxidative stress, mainly when intense or improperly managed. These effects are significantly influenced by age, gender, ethnicity, and socioeconomic status, which can affect the generation of reactive species during exercise and the body’s capacity to manage them.

The evidence suggests that moderate exercise stimulates beneficial adaptive responses through mechanisms like the activation of the NRF2 and ARE pathways, improving antioxidant defenses and cellular function. Conversely, excessive physical activity can lead to harmful levels of oxidative stress, suggesting the need for well-balanced exercise regimes. Additionally, the interaction between exercise-induced ROS and RNS production and the use of dietary antioxidants presents a complex but critical area for developing effective interventions.

Clinically, these insights advocate for personalized exercise programs that consider individual demographic factors and baseline oxidative stress levels to maximize the health benefits and minimize potential harm. Healthcare providers should consider these factors when prescribing exercise regimes and recommend dietary adjustments that enhance the body’s natural antioxidant capacity. Further research is needed to develop more precise guidelines tailored to individual needs, improving the overall effectiveness of physical activity as a preventive and therapeutic tool in managing oxidative and nitrosative stress. This approach aligns with the initial purpose of our study, aiming to enhance the understanding and management of exercise-induced biochemical changes across diverse populations.

## Figures and Tables

**Figure 1 antioxidants-13-00573-f001:**
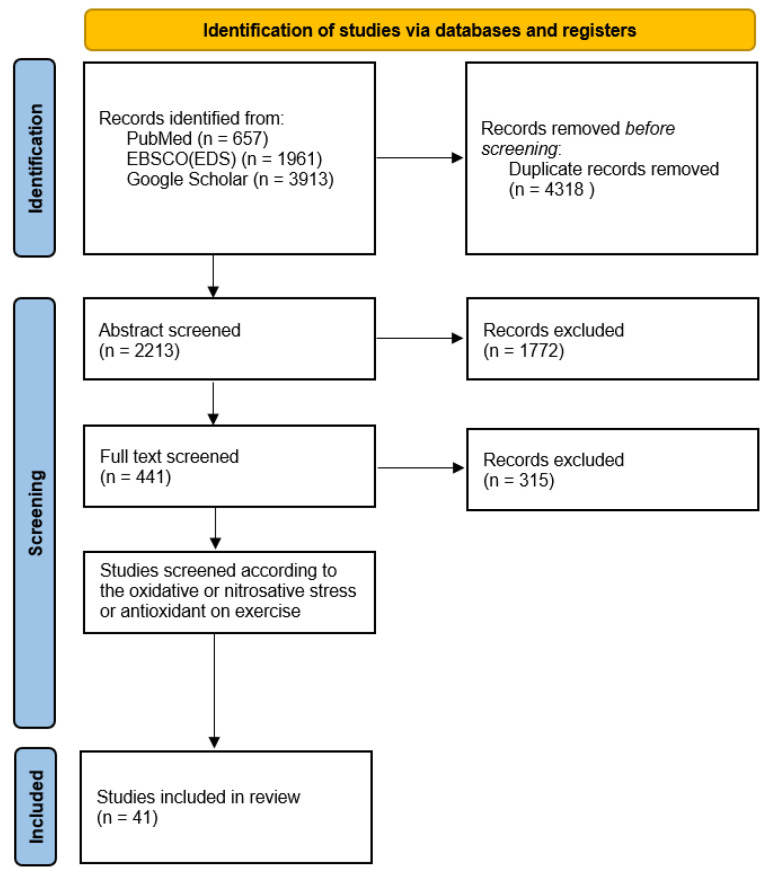
Search strategy.

**Table 1 antioxidants-13-00573-t001:** Quality assessment of the reviewed studies.

Authors and Publishing Year	Clarity of Objectives	Coherence of Arguments	Comprehensiveness of Literature	Critical Analysis of Findings	Overall Quality Rating
Powers et al. (2007) [19]	Clearly defined	Highly coherent	Comprehensive coverage	Exceptionally insightful	Very High
Huertas et al. (2017) [20]	Adequately defined	Mostly coherent	Sufficient coverage	Basic analysis	High
Gomez-Cabrera et al. (2005) [21]	Clearly defined	Highly coherent	Broad coverage	Basic analysis	High
Tiidus. (1998) [22]	Adequately defined	Some inconsistencies	Broad coverage	Basic analysis	Moderate
Peake et al. (2017) [23]	Clearly defined	Highly coherent	Comprehensive coverage	Exceptionally insightful	Very High
Su et al. (2009) [24]	Adequately defined	Mostly coherent	Broad coverage	Basic analysis	Moderate
Di Meo et al. (2019) [25]	Clearly defined	Highly coherent	Comprehensive coverage	Thorough and insightful	Very High
Cho et al. (2022) [26]	Adequately defined	Mostly coherent	Broad coverage	Basic analysis	High
Alfadda et al. (2012) [27]	Adequately defined	Mostly coherent	Broad coverage	Basic analysis	High
Gomes et al. (2012) [28]	Adequately defined	Mostly coherent	Broad coverage	Detailed analysis	High
Clemente-Suárez et al. (2023) [29]	Clearly defined	Highly coherent	Comprehensive coverage	Thorough and insightful	Very High
Black et al. (2008) [30]	Clearly defined	Highly coherent	Broad coverage	Detailed analysis	High
Nosarev et al. (2015) [31]	Adequately defined	Mostly coherent	Broad coverage	Basic analysis	High
Lim et al. (2022) [32]	Clearly defined	Highly coherent	Comprehensive coverage	Exceptionally insightful	Very High
Di Meo et al. (2016) [33]	Clearly defined	Highly coherent	Comprehensive coverage	Thorough and insightful	Very High
Yoon et al. (2021) [34]	Clearly defined	Mostly coherent	Extensive coverage	Thorough and insightful	High
Radi. (2018) [35]	Adequately defined	Mostly coherent	Broad coverage	Detailed analysis	High
Sahebnasagh et al. (2022) [36]	Clearly defined	Highly coherent	Comprehensive coverage	Thorough and insightful	Very High
Nitti et al. (2022) [37]	Adequately defined	Mostly coherent	Broad coverage	Detailed analysis	High
Powers et al. (2020) [38]	Adequately defined	Mostly coherent	Broad coverage	Detailed analysis	High
Awang Daud et al. (2022) [39]	Adequately defined	Mostly coherent	Broad coverage	Basic analysis	High
Baker et al. (2022) [40]	Adequately defined	Mostly coherent	Broad coverage	Basic analysis	Moderate
Rahaman et al. (2023) [41]	Adequately defined	Mostly coherent	Broad coverage	Basic analysis	Moderate
Rodella et al. (2023) [42]	Clearly defined	Highly coherent	Comprehensive coverage	Thorough and insightful	Very High
Braakhuis. (2012) [43]	Adequately defined	Mostly coherent	Broad coverage	Detailed analysis	High
Dewangga et al. (2023) [44]	Adequately defined	Mostly coherent	Broad coverage	Basic analysis	High
Goldfarb et al. (2022) [45]	Adequately defined	Mostly coherent	Broad coverage	Basic analysis	Moderate
Cheng et al. (2020) [46]	Adequately defined	Mostly coherent	Broad coverage	Detailed analysis	High
Pałka et al. (2023) [47]	Adequately defined	Mostly coherent	Broad coverage	Detailed analysis	High
Stookey et al. (2020) [48]	Adequately defined	Mostly coherent	Broad coverage	Detailed analysis	High
Zeng et al. (2021) [49]	Clearly defined	Highly coherent	Extensive coverage	Detailed analysis	High
Nocella et al. (2019) [50]	Adequately defined	Mostly coherent	Broad coverage	Basic analysis	Moderate
Kurutas. (2016) [51]	Clearly defined	Highly coherent	Comprehensive coverage	Thorough and insightful	Very High
Bojarczuk et al. (2022) [52]	Adequately defined	Mostly coherent	Broad coverage	Basic analysis	Moderate
Rogers et al. (2023) [53]	Adequately defined	Mostly coherent	Broad coverage	Basic analysis	Moderate
Mohd Daud et al. (2023) [54]	Clearly defined	Highly coherent	Extensive coverage	Detailed analysis	High
Yanling et al. (2022) [55]	Adequately defined	Mostly coherent	Broad coverage	Detailed analysis	High
Torre et al. (2021) [56]	Adequately defined	Mostly coherent	Broad coverage	Detailed analysis	High
Bowtell et al. (2019) [57]	Clearly defined	Highly coherent	Comprehensive coverage	Exceptionally insightful	Very High
Zhang et al. (2023) [58]	Clearly defined	Highly coherent	Extensive coverage	Detailed analysis	High
Bjørklund et al. (2023) [59]	Adequately defined	Some inconsistencies	Broad coverage	Basic analysis	Moderate

Column Descriptions: Authors and Publishing Year—A unique identifier for each review; Clarity of Objectives—Assesses whether the objectives of the review are clearly defined and specific; Coherence of Arguments—Evaluates the logical consistency and flow of the arguments presented in the review; Comprehensiveness of Literature—Measures the scope and breadth of the literature referenced in the review, considering whether it adequately represents the field being discussed; Critical Analysis of Findings—Examines the depth and thoughtfulness of the review’s analysis concerning the findings and their implications; Overall Quality Rating—A summary rating based on the assessment of the other four criteria, typically rated as Low, Moderate, High, or Very High.

**Table 2 antioxidants-13-00573-t002:** Detailed overview of exercise-induced oxidative stress mechanisms and antioxidant responses.

Study Description	Participant Demographics	Study Design and Intervention Details	Examined Result	Conclusions	Practical Implications	Typical Ranking of Different Types of Studies
Exercise-induced oxidative stress: Cellular mechanisms and impact on muscle force production; Powers et al. (2008)	None (narrative review).	Narrative review.	Oxidants can modulate cell signaling pathways and gene expression. High levels of reactive oxygen species lead to muscle weakness.	Exercise-induced oxidative stress affects muscle force production. Reactive oxygen species influence skeletal muscle contractile properties.	Oxidants affect muscle force; balance is crucial for performance. The research aims to protect muscles from oxidant-induced dysfunction.	[19]: D
Antioxidant effect of exercise: Exploring the role of the mitochondrial complex I superassembly; Huertas et al. (2017)	Male Wistar rats were used in the study. No human participants were involved in the research.	Male Wistar rats trained or sedentary for ten weeks. Blood samples collected, gastrocnemius muscle isolated for assessment.	Exercise induced CI superassembly in gastrocnemius, decreasing lipid peroxidation. No significant decrease in plasma PC due to high variability.	Exercise induced assembly of mitochondrial CI into supercomplexes. Systemic decrease in lipid peroxidation observed after exercise.	Exercise-induced superassembly of mitochondrial CI improves redox homeostasis. Further research is needed for implications in mitochondrial medicine and sport science.	[20]: B
Decreasing xanthine oxidase-mediated oxidative stress prevents useful cellular adaptations to exercise in rats; Gomez-Cabrera et al. (2005)	Twenty male Wistar rats were randomly divided into three groups: rest (*n* = 5), exercised (*n* = 5), and exercised but pretreated with 32 mg kg^−1^ of allopurinol by intra-peritoneal (i.p.) injection (*n* = 5) (Viña et al. 2000).	Allopurinol pretreatment in rats before exhaustive exercise. Three groups: rest, exercised, exercised with allopurinol pretreatment.	RONS in exercise activate MAPKs, leading to cell adaptations. Antioxidants may hinder beneficial cell adaptations induced by exercise.	Antioxidants before exercise may prevent useful cellular adaptations. Reactive oxygen or nitrogen species (RONS) act as signals during exercise.	Antioxidants before exercise may hinder beneficial cellular adaptations. Reactive oxygen or nitrogen species (RONS) act as signals in exercise.	[21]: B
Radical species in inflammation and overtraining; Tiidus et al. (1998)	None (narrative review).	Narrative review.	Exercise enhances HOCl formation in circulating neutrophils. Neutrophil HOCl generation may inhibit tissue antiproteases.	Role of reactive oxygen species in exercise-induced muscle damage. Training-induced modifications in muscle antioxidants affect inflammatory response.	Exercise may enhance HOCl formation in neutrophils. Neutrophil-generated reactive oxygen species play a role in muscle damage.	[22]: D
Muscle damage and inflammation during recovery from exercise; Peake et al. (2017)	None (narrative review).	Narrative review.	The paper discusses the effects of exercise-induced muscle damage on recovery. It highlights the importance of inflammation in muscle repair and regeneration.	Inflammation is integral to muscle repair and regeneration. More research needed on interventions for exercise-induced muscle damage.	Understanding factors influencing muscle recovery and inflammation after exercise. Investigating interventions for reducing exercise-induced muscle damage symptoms.	[23]: D
Direct detection of the oxygen rebound intermediates, ferrylMb and NO2, in the reaction of metmyoglobin with peroxynitrite; Su et al. (2009)	The focus of the sources is on the biochemical analysis of the reaction between metmyoglobin and peroxynitrite, involving methods like stopped-flow spectrophotometry and mass spectrometry, rather than on study participants or demographic data.	Stopped-flow spectrophotometry. Mass spectrometer controlled by Xcalibur software version 2.0.	FerrylMb and freely diffusing NO2 are detected as products. Ratio of in-cage rebound to cage escape is approximately 10.	FerrylMb and freely diffusing NO2 are detected intermediates. MetMb/PN interaction leads to nitration and oxygen rebound scenarios.	Understanding heme protein reactions with peroxynitrite and nitric oxide. Insights into nitrosative protein damage and cell injuries.	[24]: C
Mediators of physical activity protection against ROS-linked skeletal muscle damage; Di Meo et al. (2019)	Age groups: young, adult, old rats and mice. Training effects on antioxidant enzymes in skeletal muscle.	Cell culture study using C2C12 skeletal muscle cells. Literature review on sources of ROS production and redox signaling pathways.	Regular physical activity reduces risk of various diseases. ROS activation leads to upregulation of endogenous antioxidant defenses.	Exercise-induced ROS activate Nrf2, enhancing antioxidant defenses. Redox signaling pathways influence muscle adaptation to oxidative stress.	Understanding ROS role in muscle adaptation to exercise. Identifying redox signaling pathways for muscle phenotype changes.	[25]: D
Impact of exercise intensity on systemic oxidative stress, inflammatory responses, and sirtuin levels in healthy male volunteers; Cho et al. (2022)	Twenty healthy, untrained male volunteers. No regular exercise, injuries, diseases, or smoking history.	Treadmill running at different intensities Blood samples obtained pre-, post-, and 1 h post-exercise.	Exercise intensity affects oxidative stress, inflammatory responses, and SIRT levels. High-intensity exercise leads to higher oxidative stress and IL-6 levels.	Exercise intensity affects oxidative stress, inflammatory responses, and SIRT levels. SIRT-1 and SIRT-3 levels increase with acute exercise regardless of intensity.	Consider exercise intensity for oxidative stress and inflammatory response modulation. SIRT-1 and SIRT-3 levels increase regardless of exercise intensity.	[26]: B
Reactive oxygen species in health and disease; Alfadda et al. (2012)	None (narrative review).	Review of research on ROS roles in health and disease. Discussion on ROS production in endoplasmic reticulum and mitochondria.	ROS play key roles in health and disease. Endoplasmic reticulum and mitochondria are involved in ROS production.	ROS play key roles in health and disease. Endoplasmic reticulum and mitochondria are involved in ROS production.	Redox regulation plays a crucial role in health and disease. Antioxidants may have potential therapeutic value.	[27]: D
Oxidants, antioxidants, and the beneficial roles of exercise-induced production of reactive species; Gomes et al. (2012)	None (narrative review).	Exercise-induced reactive species production mechanisms: mitochondrial electron transport chain. Other mechanisms: ischemia/reperfusion, xanthine oxidase activation, inflammatory response, catecholamines.	Reactive species produced during exercise have beneficial effects on exercise adaptation. These effects include angiogenesis, mitochondria biogenesis, and muscle hypertrophy.	Reactive species produced during exercise have beneficial effects on exercise adaptation. Chronic exercise upregulates the body’s antioxidant defense mechanism to minimize oxidative stress.	Exercise-induced reactive species lead to beneficial training adaptations. Chronic exercise enhances body’s antioxidant defense mechanism.	[28]: D
Antioxidants and sports performance; Clemente-Suárez et al. (2023)	Athletes and recreational exercisers. No specific demographic details provided.	Analyzing reactive oxygen species and antioxidant response in sports performance. Discussing antioxidants supplementation strategies for enhancing physical and mental well-being.	ROS critical for training adaptation in resistance training. Antioxidants like vitamin C, E, and selenium enhance performance.	ROS plays a critical role in training adaptation induced by resistance training. Micronutrients and antioxidants enhance recovery and sports performance.	Antioxidants like vitamin C, E enhance recovery and performance. Micronutrients counteract free radicals, aiding oxidative stress reduction.	[29]: D

The level of evidence of each reference is graded as B: Randomized controlled trials (RCTs); C: Cohort studies, case–control studies, cross-sectional surveys, case studies, and/or observational studies; D: Review or evidence insufficient for categories A to C.

**Table 3 antioxidants-13-00573-t003:** Detailed insights into exercise-induced nitrosative stress: mechanisms, benefits, and regulatory strategies.

Study Description	Participant Demographics	Study Design and Intervention Details	Examined Result	Conclusions	Practical Implications	Typical Ranking of Different Types of Studies
Exercise prevents age-related decline in nitric-oxide-mediated vasodilator function in cutaneous microvessels; Black et al. (2008)	Young, older sedentary, older fit subjects participated in the study. Older sedentary group underwent exercise training for 12 and 24 weeks.	Heating skin to 42 °C, infusing Ringer solution or l-NAME Infusing incremental doses of ACh with or without l-NAME	Exercise prevents age-related decline in microvascular NO-mediated vasodilator function. Physiological and pharmacological NO-mediated microvascular responses improved with exercise.	Exercise prevents age-related decline in microvascular NO-mediated vasodilator function. Higher NO levels from exercise have anti-atherogenic benefits.	Exercise prevents age-related decline in microvascular NO-mediated vasodilator function. Higher NO levels from exercise may prevent microvascular dysfunction in humans.	[30]: C
Exercise and NO production: relevance and implications in the cardiopulmonary system; Nosarev et al. (2015)	None (narrative review).	Review of NO sources, NOS forms, and exercise effects. Analysis of NO modulation in health and disease contexts.	Exercise increases NO bioavailability, antioxidant defense, and cardiovascular function. Physical activity is beneficial for treating cardiovascular diseases.	Exercise improves cardiovascular function through increased NO bioavailability. Regular physical activity is beneficial for treating cardiovascular diseases.	Exercise improves cardiovascular function through increased NO bioavailability. Regular physical activity is beneficial for treating cardiovascular diseases.	[31]: D
The effects of exercise training on mitochondrial function in cardiovascular diseases: a systematic review and meta-analysis; Lim et al. (2022)	CVD patients. No specific demographic details provided.	Systematic review using PubMed, Scopus, Web of Science databases. Meta-analysis on exercise training effect on mitochondrial function in CVD.	Exercise training improves mitochondrial oxidative capacity in cardiovascular disease patients. Results on mitochondrial morphology, biogenesis, dynamics, antioxidant capacity, and quality are inconclusive.	Exercise training improves mitochondrial oxidative capacity in cardiovascular disease patients. Further research is needed to understand the effects on other mitochondrial parameters.	Exercise improves mitochondrial oxidative capacity in cardiovascular disease patients. Further research needed on mitochondrial quantity and quality modifications.	[32]: A
Role of ROS and RNS sources in physiological and pathological conditions; Di Meo et al. (2016)	None (narrative review).	Study on ROS and RNS sources in physiological conditions. Investigation of interplay between cellular ROS sources.	ROS and RNS have dual role in oxidative damage and signaling. Mitochondria and other cellular sources contribute to tissue damage and survival.	ROS and RNS play dual role in oxidative damage and signaling. Mitochondria and other cellular sources contribute to tissue oxidative stress.	ROS sources contribute to tissue damage and survival mechanisms. Mitochondria interplay with other cellular ROS sources in tissues.	[33]: D
Nitrosative stress and human disease: therapeutic potential of denitrosylation; Yoon et al. (2021)	None (narrative review).	S-nitrosylation and transnitrosylation pathways. Denitrosylation by glutathione and thioredoxin systems.	The paper provides an introduction to nitrosation and denitrosylation processes. The paper discusses nitrosation-associated human diseases and a possible denitrosylation strategy.	The article emphasizes the importance of understanding protein nitrosation and its implications in human health.	Denitrosylation as a therapeutic strategy. Understanding nitrosation in human diseases for targeted interventions.	[34]: D
Oxygen radicals, nitric oxide, and peroxynitrite: Redox pathways in molecular medicine; Rady et al. (2018)	None (narrative review).	Biochemical characterization, identification, and quantitation of peroxynitrite. Relationship of nitric oxide with redox intermediates and metabolism.	Peroxynitrite is a cytotoxic effector in disease processes. Nitric oxide interacts with superoxide radical to form peroxynitrite.	Oxygen radicals and peroxynitrite play a role in cell degeneration and death. Peroxynitrite is a cytotoxic effector against pathogens.	Reveals role of peroxynitrite in disease processes. Peroxynitrite acts as endogenous cytotoxin and cytotoxic effector.	[35]: D

The level of evidence of each reference is graded as A: Systematic reviews and meta-analyses; C: Cohort studies, case–control studies, cross-sectional surveys, case studies, and/or observational studies; D: Review or evidence insufficient for categories A to C.

**Table 4 antioxidants-13-00573-t004:** Comprehensive strategies for optimizing health benefits and mitigating.

Study Description	Participant Demographics	Study Design and Intervention Details	Examined Result	Conclusions	Practical Implications	Typical Ranking of Different Types of Studies
Neurohormetic phytochemicals in the pathogenesis of neurodegenerative diseases; Sahebnasagh et al. (2022)	None (narrative review).	Evaluation of neurohormetic dose-response concept. Review of neurohormetic phytochemicals for neurodegenerative diseases.	Herbal compounds like curcumin, quercetin, and fisetin show neuroprotective effects. Phytochemicals activate neurohormesis, potentially treating neurodegenerative diseases.	Neurohormesis has potential benefits in neurodegenerative and other neurological disorders. Phytochemicals have shown therapeutic effects in neurodegenerative diseases at low doses.	Herbal phytochemicals show potential in managing neurodegenerative diseases. Hormesis concept can slow onset and reduce damage in diseases.	[36]: D
Hormesis and oxidative distress: pathophysiology of reactive oxygen species and the open question of antioxidant modulation and supplementation; Nitti et al. (2022)	None (narrative review).	Antioxidant modulation. Endogenous antioxidant depletion.	Antioxidant supplementation shows conflicting or incomplete results in disease prevention. Endogenous antioxidant depletion may enhance susceptibility to anticancer therapies.	Antioxidant supplementation may not prevent chronic pathologies effectively. Depletion of endogenous antioxidants can enhance anticancer therapy efficacy.	Antioxidant modulation may enhance anticancer therapy effectiveness. Deep investigation of redox balance for more effective therapies.	[37]: D
Exercise-induced oxidative stress: friend or foe?; Powers et al. (2020)	None (narrative review).	Exercise protocols. Endogenous antioxidant system.	The paper discusses the effects of physical exercise on oxidative stress. It suggests that specific exercise protocols can control ROS production and induce adaptation.	Physical exercise can be recommended for maintaining a healthy lifestyle. Specific exercise protocols can disperse the transient effect of ROS and prevent local oxidative damage.	Exercise can control ROS production, inducing adaptation for health benefits. Specific exercise protocols can disperse ROS effects and maintain redox homeostasis.	[38]: D
Oxidative stress and antioxidant enzymes activity after cycling at different intensity and duration; Awang Daud et al. (2022)	Twenty-five sedentary adults participated in the study. No specific demographic details provided beyond sedentary adults.	A crossover trial was employed in this study.	Cycling increased oxidative stress and antioxidant activities in sedentary adults. Different responses observed based on intensity and duration of exercise.	Exercise intensity should not exceed 70% VO2pk for sedentary adults. Cycling increased oxidative stress and antioxidant activities with varying responses.	Exercise intensity for sedentary adults should not exceed 70% VO2pk. Different exercise durations affect oxidative stress and antioxidant enzyme activities.	[39]: C
High-intensity exercise performance and muscle damage. A role for free radicals; Baker et al. (2022)	None (narrative review).	The article discusses the impact of high-intensity exercise on muscle damage and the role of free radicals in this process.	Muscle damage and oxidative stress responses to high intensity exercise. Role of antioxidants in recovery process and reducing oxidative stress damage.	Muscle damage and oxidative stress responses to high intensity exercise. Role of antioxidants in recovery process post-exercise.	Recovery strategies after high intensity exercise. Role of antioxidants in reducing oxidative stress damage.	[40]: D
Natural antioxidants from some fruits, seeds, foods, natural products, and associated health benefits: An update; Kambizi et al. (2023)	None (narrative review).	The information was collected from scientific databases, professional websites, and traditional medicine books. The paper included analysis and in-depth discussion of studies on phytochemistry and pharmacological effects.	The paper highlights the pharmacological relevance of antioxidants in natural sources. Natural antioxidants have different health-promoting effects and can be used as functional foods.	Natural antioxidants have health-promoting effects. Functional foods with high antioxidant potential are beneficial.	Antioxidants can be beneficial in preventing diseases such as heart disease and cancer, with exogenous antioxidants derived from both natural and synthetic sources.	[41]: D
Antioxidant nutraceutical strategies in the prevention of oxidative stress related eye diseases; Rodella et al. (2023)	None (narrative review).	Discussion on antioxidant nutraceutical molecules in managing radicals in the eye. Presentation of different antioxidant molecules in anatomical compartments of the eye.	Mixed/inconclusive results on antioxidant supplementation efficacy. Need for future research on antioxidant potential and new strategies.	The delicate balance between reactive oxygen species and antioxidants in the eye is crucial for eye health and preventing oxidative stress-related disorders. Antioxidant molecules, both endogenously produced and obtained through the diet, play a role in maintaining this balance.	Antioxidant nutraceuticals may help prevent oxidative stress-related eye diseases. Studies on antioxidant supplementation efficacy require further research for validation.	[42]: D
Effect of vitamin C supplements on physical performance; Braakhuis et al. (2012)	None (narrative review).	Studies were sourced via Google Scholar and reference lists in related articles and books. Inclusion criteria required high-intensity maximal performance tests and vitamin C supplementation.	Vitamin C supplements (>1 g/day) may impair physical performance by reducing mitochondrial biogenesis. Lower doses (~0.2 g/day from fruits/vegetables) can reduce oxidative stress without hindering training adaptations.	Vitamin C supplements in doses >1 g·d (−1) impair sport performance. Doses of ∼0.2 g·d (−1) of vitamin C from fruits and vegetables reduce oxidative stress without impairing training adaptations.	Vitamin C supplements >1 g/day may impair sport performance. Consuming 0.2 g/day through fruits/vegetables reduces oxidative stress without impairing training.	[43]: D
The differences frequency of weekly physical exercise in antioxidant serum levels and muscle damage; Dewangga et al. (2023)	Male Wistar rats were used as participants. Sample size: 24 rats divided into four groups.	Experimental design with male Wistar rats. Serum antioxidant and muscle damage biomarkers analyzed.	Exercise 4 times/daily reduces antioxidants, increases muscle damage. SOD levels increased with exercise twice a week.	Exercise needs sufficient recovery time to prevent muscle damage. Too frequent exercise reduces serum antioxidants and increases muscle damage.	Proper recovery time in exercise crucial for antioxidant levels. Excessive exercise can lead to increased muscle tissue damage.	[44]: B
Exercise-induced oxidative stress during exercise and recovery in hot condition; Goldfarb et al. (2022)	None (narrative review).	Exercise condition (70–75% max power output). Rest condition in a hot environment (40 °C and 30% relative humidity).	Exercise did not induce oxidative stress as hypothesized. Environmental conditions and small sample size impacted study results.	Exercise did not induce oxidative stress as hypothesized. Environmental conditions and small sample size impacted study outcomes.	For athletes and active individuals, a balanced diet rich in natural antioxidants from fruits, vegetables, and whole grains is preferable to antioxidant supplements, which may not offer added benefits and could inhibit exercise-induced health gains.	[45]: D
Intramuscular mechanisms of overtraining; Cheng et al. (2020)	None (narrative review).	Assess inflammation and OTS, oxidative stress and OTS. Interventions for mitigating OTS in skeletal muscle.	Skeletal muscle weakness in overtraining syndrome linked to ROS and inflammation. Potential interventions to mitigate overtraining syndrome discussed in the paper.	ROS and inflammation likely contribute to overtraining syndrome in muscles. Interventions can help mitigate overtraining syndrome in skeletal muscles.	Understanding mechanisms of overtraining in skeletal muscle for athlete performance. Potential interventions to mitigate overtraining syndrome in skeletal muscle.	[46]: D
Effects of different hydration strategies in young men during prolonged exercise at elevated ambient temperatures on pro-oxidative and antioxidant status markers, muscle damage, and inflammatory status; Pałka et al. (2023)	Twelve healthy men with average aerobic capacity participated. Study conducted on young men during prolonged exercise at high temperatures.	Hydration strategies: no hydration, water, isotonic drinks. Exercise test on cycle ergometers to determine individual relative loads.	Isotonic drinks most effective in preventing muscle cell damage. No significant difference in oxidative status based on hydration strategy.	Isotonic drinks protect muscle cells effectively during exercise. Different hydration strategies impact plasma volume and muscle cell protection.	Isotonic drinks recommended for physical exertion in high environmental temperatures. Most effective in protecting muscle cells during exercise.	[47]: C
Underhydration is associated with obesity, chronic diseases, and death within 3 to 6 years in the U.S. population aged 51–70 years; Stookey et al. (2020)	Adults aged 51–70 years in the U.S. NHANES 2009–2012, sample size *n* = 1200.	National Health and Nutrition Examination Survey (NHANES). National Center for Health Statistics linked mortality files.	Over 65% of adults aged 51–70 in the US do not meet hydration criteria. Underhydration is associated with obesity, chronic diseases, and death within 3 to 6 years.	Underhydration is associated with obesity, chronic diseases, and death within 3 to 6 years in the U.S. population aged 51–70 years. Meeting hydration criteria is associated with zero chronic disease deaths.	Underhydration linked to obesity, chronic diseases, and increased mortality risk. Meeting hydration criteria associated with zero chronic disease deaths.	[48]:C

The level of evidence of each reference is graded as B: Randomized controlled trials (RCTs); C: Cohort studies, case–control studies, cross-sectional surveys, case studies, and/or observational studies; D: Review or evidence insufficient for categories A to C.

**Table 5 antioxidants-13-00573-t005:** Enhanced understanding of the role of dietary antioxidants in exercise-induced oxidative stress mitigation.

Study Description	Participant Demographics	Study Design and Intervention Details	Examined Result	Conclusions	Practical Implications	Typical Ranking of Different Types of Studies
Effects of dietary strategies on exercise-induced oxidative stress: A narrative review of human studies; Zeng et al. (2021)	None (narrative review).	Searched electronic databases: PubMed, Scope, Web of Science. Included 28 studies in the narrative review.	Majority of studies show favorable effects of whole dietary strategies. Further systematically designed studies are needed to strengthen evidence.	Whole dietary strategies have favorable effects on exercise-induced oxidative stress. Further studies are needed to strengthen the evidence.	Whole diets rich in antioxidants may reduce exercise-induced oxidative stress. Further studies needed for stronger evidence on dietary strategies.	[49]: D
Impairment between oxidant and antioxidant systems: short- and long-term implications for athletes’ health; Nocella et al. (2019)	None (narrative review).	Exercise training practice. Antioxidant supplementations for preventing oxidative damages.	Exercise training can induce oxidative stress and muscle damage. Antioxidant supplementation may help prevent oxidative damages in athletes.	Exercise training impacts oxidant-antioxidant balance in athletes. Antioxidant supplementation may prevent oxidative damage in athletes.	Exercise training can impact oxidant-antioxidant balance in athletes. Antioxidant supplementation may help prevent oxidative damage in athletes.	[50]: D
The importance of antioxidants which play the role in cellular response against oxidative/nitrosative stress: Current state; Kurutas et al. (2016)	None (narrative review).	Biomarkers of oxidative/nitrosative stress: lipid peroxidation products. Antioxidants: exogenous or endogenous molecules mitigating oxidative/nitrosative stress.	Antioxidants play a crucial role in cellular response to stress. Lipid peroxidation products used as biomarkers of oxidative damage.	Importance of antioxidants in cellular response against oxidative/nitrosative stress. Lipid peroxidation products used as biomarkers of oxidative/nitrosative stress damage.	Antioxidants crucial in combating oxidative/nitrosative stress. Lipid peroxidation products used as biomarkers for oxidative stress.	[51]: D
Polyphenol supplementation and antioxidant status in athletes: A narrative review; Bojarczuk et al. (2022)	None (narrative review).	Narrative review. Summarizes polyphenols’ bioavailability, role in oxidative stress, supplementation strategies.	Insufficient evidence to promote polyphenols specifically for athletes. More research needed on impact of antioxidants on athletes.	Use of polyphenols in athletes’ diet is debatable. Further research is needed to determine their impact.	Polyphenol supplementation impact on athletes’ antioxidant status needs further research. Existing evidence does not strongly support promoting polyphenols among athletes.	[52]: D
Vitamin C supplementation and athletic performance: A Review; Rogers et al. (2023)	None (narrative review).	Reviewed 14 randomized control trials on vitamin C and athletic performance. Vitamin C is often used with additional supplements like vitamin E.	Mixed results on the effects of vitamin C supplementation on athletic performance. Long-term high-dosage supplementation with vitamin C is not recommended.	Mixed results on the use of vitamin C supplementation for athletic performance. Long-term high-dosage supplementation with vitamin C is not recommended.	High-dose vitamin C supplementation is not recommended for athletes. Athletes should obtain antioxidants through a nutrient-rich diet instead.	[53]: D
Pure juice supplementation: Its effect on muscle recovery and sports performance; Mohd Daud et al. (2023)	None (narrative review).	The paper reviews previous literature on the effect of fruit juice supplementation on muscle recovery and sports performance. The paper discusses the use of natural-based fruit-derived antioxidants as a nutritional strategy.	Fruit juice supplementation aids muscle recovery and sports performance. Antioxidant-rich fruits protect muscle cells from harmful reactive oxygen species.	Fruit juice supplementation aids muscle recovery and sports performance. Antioxidant-rich fruit juices reduce oxidative stress and improve performance.	Fruit juice supplementation aids muscle recovery and sports performance in athletes. Natural antioxidants in fruit juices reduce oxidative stress and DOMS.	[54]: D
The vitamin E consumption effect on muscle damage and oxidative stress: A systematic review and meta-analysis of randomized controlled trials; Yanling et al. (2022)	Thirteen studies included in systematic review and meta-analysis.	Meta-analyses of randomized controlled trials. Placebo-controlled RCTs evaluating combined effects of vitamins C and E.	Vitamins C and E reduce lipid peroxidation and IL-6 levels. Supplementation does not affect muscle soreness or muscle strength.	Vitamins C and E reduce oxidative stress and inflammation post-exercise. No significant effect on muscle soreness and muscle strength observed.	Vitamins C and E reduce oxidative stress and inflammation post-exercise. No significant impact on muscle soreness or muscle strength observed.	[55]: A
Supplementation with vitamins C and E and exercise-induced delayed-onset muscle soreness: A systematic review; Torre et al. (2021)	280 participants: 230 men, 50 women. Healthy individuals aged 16–30 with varying physical activity levels.	Systematic reviewed studies on vitamin C and E supplementation for muscle soreness. Searched databases for randomized control trials on antioxidant intake.	Limited evidence on vitamins C and E reducing muscle soreness. Chronic supplementation may have some benefits post-exercise.	Limited evidence on vitamins C and E reducing muscle soreness. Insufficient confirmation of antioxidant vitamins minimizing delayed-onset muscle soreness.	Insufficient evidence for vitamins C and E to reduce muscle soreness. Limited support for antioxidant vitamin intake to minimize muscle soreness.	[56]: A
Fruit-derived polyphenol supplementation for athlete recovery and performance; Bowtell et al. (2019)	Recreational active participants and trained athletes involved in studies. Faster female runners and trained male cyclists participated in research.	Quantification of plasma phenolics not included. Measurement of serum markers of oxidative damage in some studies.	Polyphenol supplementation improves exercise performance and aids muscle recovery. Studies show no adverse effects with polyphenol supplementation.	Polyphenol supplementation enhances exercise performance and aids recovery. Polyphenols reduce oxidative damage and inflammation in muscle recovery process.	Polyphenol supplementation can improve performance and aid recovery in athletes. Fruit-derived polyphenols assist in muscle recovery and reduce soreness.	[57]: D
Selenium and selenoproteins in health; Zhang et al. (2023)	Selenium-replete US patients. Mice fed with selenium-enriched and selenium-deficient diets.	Observational studies on selenium and cancer relationship. Intervention trials needed to verify selenium supplementation effects.	Selenium has various physiological functions in the body. Selenium supplementation has potential benefits in various diseases.	Selenium supplementation may have dual effects on health. Further studies are needed to clarify the relevance of selenium in cancer.	Selenium supplementation may have beneficial effects on cardiovascular diseases. Further intervention trials are needed to fully understand the effects of selenium supplementation.	[58]: C
Coenzyme Q10 for enhancing physical activity and extending the human life cycle; Bjørklund et al. (2023)	None (narrative review).	Evaluation of CoQ10 functions in human health. Assessing benefits of CoQ10 supplementation in various health conditions.	CoQ10 supplementation beneficial for cardiovascular disease, less effective for statin-related symptoms. Further studies needed to clarify benefits in various health conditions.	Further studies with more patients are needed to clarify the benefits of CoQ10 therapy. CoQ10 supplementation may not add value to statin-associated muscular symptoms.	Further studies needed to clarify CoQ10 benefits in health conditions. CoQ10 supplementation may not benefit statin-associated muscular symptoms.	[59]: D

The level of evidence of each reference is graded as A: Systematic reviews and meta-analyses;C: Cohort studies, case–control studies, cross-sectional surveys, case studies, and/or observational studies; D: Review or evidence insufficient for categories A to C.

## Data Availability

Not applicable.

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
