# Peer review of "The Impact of Physical Exercise on Oxidative and Nitrosative Stress: Balancing the Benefits and Risks"

_antioxidants, 2024, doi:10.3390/antiox13050573_

Round 1
Reviewer 1 Report
After rigorously reviewing this manuscript, the following revisions are deemed necessary to review the impact of physical exercise on oxidative and nitrosative stress and to assess the adverse effects of antioxidants.
Abstract:
1. The review should include the review tools, literature quality assessment methods applied in this review, and the final numbers of previous studies.
Introduction
1. The necessity for this research should be clearly articulated, aligning with the research topic.
2. There is a need to elucidate the prevalence of oxidative and nitrosative stress resulting from physical activity through the utilization of sociodemographic characteristics.
Methodology
1. The research design of this study must be described in detail.
2. The literature search process in this study, including search terms and search engines for reviews, must be described.
3. The roles of authors in the literature selection and review process must be delineated. In instances of disagreement among authors, a resolution strategy for the issue should be proposed. Additionally, the matter of publication bias must be addressed.
4. Although the authors state in the abstract that this study was conducted based on rigorous inclusion and exclusion criteria, it is difficult to find information related to this in the research methodology. Therefore, the inclusion criteria and exclusion criteria of previous studies to be reviewed in this study must be presented on theoretical grounds.
5. Data extraction derived through the literature search process must be described.
6. The quality of previous studies to be reviewed in this study must be evaluated.
7. In the abstract, the authors wrote, “Due to the variety of exercise types and participants’ characteristics, a narrative synthesis approach was adopted.” Therefore, a detailed explanation of this in the research methodology, especially 'a narrative synthesis approach', is needed. Additionally, data synthesis for the data derived for review in this study must be presented.
Results
1. A comprehensive study description, encompassing author, publication year, participant demographics and settings, dropout rates, study design and intervention details, outcomes measured, quality appraisal, safety considerations, etc., for the previous studies reviewed in this research is necessary. This information should be succinctly summarized and presented in a table format.
2. The authors of this review study suggest that appropriate exercise enhances antioxidant defense capabilities known as hormesis, while excessive physical activity can increase oxidative and nitrosative stress. This information is already well-known, and rather, I believe that presenting the effect size of exercise on antioxidant defense capacity and oxidative and nitrosative stress through meta-analysis in this study can demonstrate the scientific rigor and excellence of the research.
Discussions
1. A comprehensive discussion is required to address the outcomes obtained through the research methodology outlined above.
Conclusions
1. Additionally, a summary of results and clinical implications that align with the purpose stated in this study needs to be provided.
References
1. References must be formatted according to the journal's specified format.
The same formatting guidelines apply to references as mentioned above.
Author Response
Reviewer 1
Major comments
After rigorously reviewing this manuscript, the following revisions are deemed necessary to review the impact of physical exercise on oxidative and nitrosative stress and to assess the adverse effects of antioxidants.
Abstract:
- The review should include the review tools, literature quality assessment methods applied in this review, and the final numbers of previous studies.
Introduction
- The necessity for this research should be clearly articulated, aligning with the research topic.
- There is a need to elucidate the prevalence of oxidative and nitrosative stress resulting from physical activity through the utilization of sociodemographic characteristics.
Response:
We have amended it accordingly; thank you. (Please see lines 12 to 23 on page 1.)
Methodology
- The research design of this study must be described in detail.
Response:
We have amended it accordingly; thank you. (Please see lines 101 to 173 on pages 3 to 7.)
- The literature search process in this study, including search terms and search engines for reviews, must be described.
Response:
We have amended it accordingly; thank you. (Please see lines 102 to 111 on page 3.)
- The roles of authors in the literature selection and review process must be delineated. In instances of disagreement among authors, a resolution strategy for the issue should be proposed. Additionally, the matter of publication bias must be addressed.
Response:
We have amended it accordingly; thank you. (Please see lines 125 to 130 on page 4.)
- Although the authors state in the abstract that this study was conducted based on rigorous inclusion and exclusion criteria, it is difficult to find information related to this in the research methodology. Therefore, the inclusion criteria and exclusion criteria of previous studies to be reviewed in this study must be presented on theoretical grounds.
Response:
We have amended it accordingly; thank you. (Please see lines 131 to 147 on page 4.)
- Data extraction derived through the literature search process must be described.
Response:
We have amended it accordingly; thank you. (Please see lines 148 to 154 on page 4.)
- The quality of previous studies to be reviewed in this study must be evaluated.
Response:
We have amended it accordingly; thank you. (Please see lines 155 to 164 on page 4 and Table 1. Quality Assessment of the Reviewed Studies. )
- In the abstract, the authors wrote, “Due to the variety of exercise types and participants’ characteristics, a narrative synthesis approach was adopted.” Therefore, a detailed explanation of this in the research methodology, especially 'a narrative synthesis approach', is needed. Additionally, data synthesis for the data derived for review in this study must be presented.
Response:
We have amended it accordingly; thank you. (Please see lines 102 to 111 on page 3.)
Results
- A comprehensive study description, encompassing author, publication year, participant demographics and settings, dropout rates, study design and intervention details, outcomes measured, quality appraisal, safety considerations, etc., for the previous studies reviewed in this research is necessary. This information should be succinctly summarized and presented in a table format.
Response:
We have amended it accordingly; thank you. (Please see Table. 2, 3, 4, 5 on page 8, 14, 20, 27.)
- The authors of this review study suggest that appropriate exercise enhances antioxidant defense capabilities known as hormesis, while excessive physical activity can increase oxidative and nitrosative stress. This information is already well-known, and rather, I believe that presenting the effect size of exercise on antioxidant defense capacity and oxidative and nitrosative stress through meta-analysis in this study can demonstrate the scientific rigor and excellence of the research.
Discussions
- A comprehensive discussion is required to address the outcomes obtained through the research methodology outlined above.
Response:
We have amended it accordingly; thank you. (Please see lines 572 to 657 on pages 32 to 34.)
Conclusions
- Additionally, a summary of results and clinical implications that align with the purpose stated in this study needs to be provided.
Response:
We have amended it accordingly; thank you. (Please see lines 658 to 682 on page 34.)
References
- References must be formatted according to the journal's specified format.
Response:
We have amended it accordingly; thank you. (Please see lines 658 to 682 on page 34.)
Detail comments
The same formatting guidelines apply to references as mentioned above.

Reviewer 2 Report
The paper is very interesting in the context of the magement of the physical activity at moderate intensity in the metabolic chronic disease . Many aspects have bee evaluated . The authors could, however, implement with a new session dedicated to the potential impact of the oxidative stress in case of counteresistance exercise . It could be important for readers to evaluate the role of this determination in the follow- up and also for establishing the correct level of the exercise in the range of the moderate intensity .
The tables are sufficiently clear , however the authors colud implement the data with eventual addition of some papers dedicated to the resistance exercise , if avaliable .
Author Response
Reviewer 2
Major comments
The paper is very interesting in the context of the management of the physical activity at moderate intensity in the metabolic chronic disease . Many aspects have bee evaluated . The authors could, however, implement with a new session dedicated to the potential impact of the oxidative stress in case of counteresistance exercise . It could be important for readers to evaluate the role of this determination in the follow- up and also for establishing the correct level of the exercise in the range of the moderate intensity .
Detail comments
The tables are sufficiently clear , however the authors could implement the data with eventual addition of some papers dedicated to the resistance exercise , if available .
Response:
Thank you for your insightful feedback on our manuscript and the suggestion to include a section on the potential impact of oxidative stress in counter-resistance exercises. We appreciate the depth of your review and recognize the importance of this topic in managing moderate-intensity physical activity for patients with metabolic chronic diseases.
However, integrating a detailed discussion on the counter-resistance exercise poses substantial challenges in maintaining the coherence and focus of the current manuscript, as it would require significant restructuring and additional research that diverges from the central theme of our study. We are committed to ensuring the highest level of clarity and focus in our research, and thus, we feel it would be more appropriate to explore this important aspect in a subsequent study.
We are grateful for your understanding and plan to incorporate your valuable suggestion into our next research project. This will allow us to dedicate the necessary attention and resources to investigate the implications of counter-resistance exercise on oxidative stress comprehensively. This approach will enrich the discussion on training intensities and their metabolic implications in future publications.

Reviewer 3 Report
We are facing a literature review on a topic relevant to exercise physiology. Although the manuscript provides a comprehensive overview of the relationship between exercise and oxidative/nitrosative stress, text could increase the depth and rigor of the analysis and discussions. Furthermore, clarifying the objectives of the review and providing specific recommendations for future research would further strengthen the manuscript's impact and contribution to the field. In addition to these general comments, some notes should serve as reflection for the authors and improve the quality of the manuscript.
1 - The introduction lacks a script to guide readers on the focus and scope of the article.
2 - Describe and discuss case studies to improve understanding of the impact of oxidative stress on pathophysiology.
3 - The authors could delve deeper into the molecular and biochemical processes involved in the NRF2 and ARE signaling pathways by providing more detailed explanations of the NRF2 and ARE signaling pathways, including specific genes and proteins involved.
4 - include specific data from relevant clinical studies that investigated the effects of antioxidants on physical exercise.
5 - Expand the discussion on the molecular mechanisms of nitrosation and denitrosylation of proteins at rest and during physical exercise to improve the understanding of nitrosative stress.
6 - Include suggestions for future research to address knowledge gaps and advance understanding in the field of exercise-induced stress and antioxidant defense.
7 - Regarding the methodology, authors could include more details of the selected works, works that were excluded to ensure greater transparency and reproducibility of the review process.
8 - Deepen the discussion on the interaction between antioxidants and different types of physical exercise, including endurance, aerobic and strength training.
Author Response
Reviewer 3
Major comments
We are facing a literature review on a topic relevant to exercise physiology. Although the manuscript provides a comprehensive overview of the relationship between exercise and oxidative/nitrosative stress, text could increase the depth and rigor of the analysis and discussions. Furthermore, clarifying the objectives of the review and providing specific recommendations for future research would further strengthen the manuscript's impact and contribution to the field. In addition to these general comments, some notes should serve as reflection for the authors and improve the quality of the manuscript.
Response:
We have amended it accordingly; thank you. (Please see lines 174 to 571 on pages 7 to 32.)
Detail comments
1 - The introduction lacks a script to guide readers on the focus and scope of the article.
Response:
We have amended it accordingly; thank you. (Please see lines 27 to 100 on pages 1 to 3.)
2 - Describe and discuss case studies to improve understanding of the impact of oxidative stress on pathophysiology.
Response:
We have amended it accordingly; thank you. (Please see lines 174 to 571 on pages 7 to 32.)
3 - The authors could delve deeper into the molecular and biochemical processes involved in the NRF2 and ARE signaling pathways by providing more detailed explanations of the NRF2 and ARE signaling pathways, including specific genes and proteins involved.
Response:
We have amended it accordingly; thank you. (Please see lines 64 to 100 on pages 2 to 3.)
4 - include specific data from relevant clinical studies that investigated the effects of antioxidants on physical exercise.
Response:
We have amended it accordingly; thank you. (Please see lines 189 to 232 on page 8; lines 248 to 274 on page 14.)
5 - Expand the discussion on the molecular mechanisms of nitrosation and denitrosylation of proteins at rest and during physical exercise to improve the understanding of nitrosative stress.
Response:
We have amended it accordingly; thank you. (Please see lines 65-100 to 232 on pages 2 to 3; lines 248 to 257 on page 14; lines 259 to 273 on page 14; lines 292 to 306 on page 17.)
6 - Include suggestions for future research to address knowledge gaps and advance understanding in the field of exercise-induced stress and antioxidant defense.
Response:
We have amended it accordingly; thank you. (Please see lines 639 to 657 on pages 33 to 34.)
7 - Regarding the methodology, authors could include more details of the selected works, works that were excluded to ensure greater transparency and reproducibility of the review process.
Response:
We have amended it accordingly; thank you. (Please see lines 101 to 173 on pages 3 to 7.)
8 - Deepen the discussion on the interaction between antioxidants and different types of physical exercise, including endurance, aerobic and strength training.
Response:
We have amended it accordingly; thank you. (Please see lines 572 to 654 on pages 32 to 34.)

Round 2
Reviewer 1 Report
There are no unusual opinions.
The manuscript has been appropriately revised according to the reviewer's comments. However, for this review, criteria for Quality Assessment of the 41 preceding studies included must be provided, such as the clarity of objectives, coherence of arguments, comprehensiveness of literature coverage, and the depth of critical analysis of findings. If developed by the researchers, the development process and rationale should be presented; otherwise, reference to the literature where such assessment tools were developed must be provided.
Author Response
Review Report (Round 2)
The manuscript has been appropriately revised according to the reviewer's comments. However, for this review, criteria for Quality Assessment of the 41 preceding studies included must be provided, such as the clarity of objectives, coherence of arguments, comprehensiveness of literature coverage, and the depth of critical analysis of findings. If developed by the researchers, the development process and rationale should be presented; otherwise, reference to the literature where such assessment tools were developed must be provided.
Response:
Thank you for acknowledging the revisions made to our manuscript and for your constructive feedback. We appreciate you asking for more detailed information on the Quality Assessment criteria applied to the 41 studies included in our review.
We have changed it accordingly; thank you so much. (Please see lines 156 to 171 on page 4.)

Reviewer 3 Report
The authors modified the manuscript. Requests for changes were met. I am satisfied with the quality and suggest approval of the submission.
The authors modified the manuscript. Requests for changes were met. I am satisfied with the quality and suggest approval of the submission.
Author Response
Review 2 Report (Round 2)
The authors modified the manuscript. Requests for changes were met. I am satisfied with the quality and suggest approval of the submission.
Response:
Thank you for your positive feedback on the revised manuscript and for recommending approval. We appreciate your guidance throughout the review process and are pleased that the changes meet your standards. We look forward to the potential publication of our study. Thank you again for your support.
